# Ceruloplasmin-Deficient Mice Show Dysregulation of Lipid Metabolism in Liver and Adipose Tissue Reduced by a Protein Replacement

**DOI:** 10.3390/ijms24021150

**Published:** 2023-01-06

**Authors:** Sara Raia, Antonio Conti, Alan Zanardi, Barbara Ferrini, Giulia Maria Scotti, Enrica Gilberti, Giuseppe De Palma, Samuel David, Massimo Alessio

**Affiliations:** 1Proteome Biochemistry, COSR-Centre for Omics Sciences, IRCCS-San Raffaele Hospital, 20132 Milan, Italy; 2COSR-Centre for Omics Sciences, IRCCS-San Raffaele Hospital, 20132 Milan, Italy; 3Unit of Occupational Health and Industrial Hygiene, Department of Medical and Surgical Specialties, Radiological Sciences and Public Health, University of Brescia, 25121 Brescia, Italy; 4Center for Research in Neuroscience, The Research Institute of The McGill University Health Center, Montreal, QC H3G 1A4, Canada

**Keywords:** ceruloplasmin, aceruloplasminemia, steatosis, adipose tissue, adipokines, enzyme replacement therapy, iron homeostasis, infiltrating macrophages, inflammation

## Abstract

Ceruloplasmin is a ferroxidase that plays a role in iron homeostasis; its deficiency fosters inter alia iron accumulation in the liver, which expresses the soluble form of the protein secreted into the bloodstream. Ceruloplasmin is also secreted by the adipose tissue, but its role in adipocytes has been poorly investigated. We hypothesized that ceruloplasmin might have a role in iron/lipid interplay. We investigated iron/lipid dysmetabolism in the liver and adipose tissue of the ceruloplasmin-deficient mouse (CpKO) model of aceruloplasminemia and evaluated the effectiveness of ceruloplasmin replacement. We found that CpKO mice were overweight, showing adipose tissue accumulation, liver iron deposition and steatosis. In the adipose tissue of CpKO mice, iron homeostasis was not altered. Conversely, the levels of adiponectin and leptin adipokines behaved opposite to the wild-type. Increased macrophage infiltration was observed in adipose tissue and liver of CpKO mice, indicating tissue inflammation. The treatment of CpKO mice with ceruloplasmin limited liver iron accumulation and steatosis without normalizing the expression of iron homeostasis-related proteins. In the CpKO mice, the protein replacement limited macrophage infiltration in both adipose and hepatic tissues reduced the level of serum triglycerides, and partially recovered adipokines levels in the adipose tissue. These results underline the link between iron and lipid dysmetabolism in ceruloplasmin-deficient mice, suggesting that ceruloplasmin in adipose tissue has an anti-inflammatory role rather than a role in iron homeostasis. Furthermore, these data also indicate that ceruloplasmin replacement therapy may be effective at a systemic level.

## 1. Introduction

Ceruloplasmin (Cp) is an extracellular multi-copper ferroxidase that plays a role in iron homeostasis by promoting both the cellular iron efflux through membrane stabilization of the iron exporter ferroportin (Fpn) and the incorporation of the oxidized iron onto transferrin (Tf) [1,2,3,4,5]. In addition to this major physiological role, several other less-defined functions have been ascribed to Cp. These include nitric oxide- and amine-oxidase, copper transport, anti-oxidant and anti-inflammatory activities [1,6,7,8,9]. However, the major function of Cp based on findings in both humans with Cp gene deficiency and CpKO mice show that the lack of Cp ferroxidase activity leads to a decrease in Tf saturation, iron-restricted erythropoiesis, and to iron accumulation in the liver, pancreas, retina and brain [10]. Indeed, the rare genetic disease aceruloplasminemia (Acp), associated with the absence of Cp or the presence of an inactive form of the protein, shows microcytic anemia and complications due to iron deposition, such as retinal degeneration, diabetes and neurodegeneration in the brain [10,11]. Despite the dramatic iron accumulation, other than a general inflammatory status in the CpKO mice and steatosis in some Acp patients, no overt clinical consequences have been reported for the liver in the absence of Cp function [12,13,14,15].

There are two isoforms of Cp, a secreted and a membrane-bound form expressed in various tissues [1,3,16]. At the systemic level, Cp is mainly expressed by the liver, the primary regulator of iron metabolism, as the secreted form is released into the bloodstream [1]. Cp is also expressed by several other cell types, including the adipocytes that release it as an adipokine [12,17,18]. A dysregulated interplay between iron and lipids metabolism is a hallmark of several metabolic diseases and is characterized by altered levels of key players in the regulation of iron and lipid metabolism, like hepcidin secreted by the liver, or adipokines released by adipose tissue (e.g., adiponectin, leptin), which can be in turn regulated by iron level and contribute to the control of hepcidin secretion by adipocytes [19,20,21,22]. Thus, the interaction between hepatocytes and adipocytes is regulated by feedback mechanisms linking iron, iron homeostasis-related proteins and adipokines. Although Cp is secreted by both the liver and adipose tissue and counteracts hepcidin function in the regulation of Fpn [2,23], its role in the interplay between hepatocytes and adipocytes, has been poorly investigated, as well as the role of adipocytes in the pathogenic mechanisms of Acp [12].

Exploiting the CpKO mouse model, we show that in the absence of this molecule, which is involved in iron homeostasis, in the liver and adipose tissue, there are also alterations in lipids metabolism and in the level of some lipid metabolism-related molecules. This work unravels the metabolic interplay occurring between liver and adipose tissue in Cp-deficient mice and suggests an anti-inflammatory role of ceruloplasmin in adipocytes that might contribute to the Acp pathology. The involvement of Cp in leading these effects is underlined by the administration of purified Cp, which ameliorates the iron and lipid dysmetabolism in the liver and adipose tissue of CpKO mice. In turn, as reported for the neurological symptoms [24], Cp replacement therapy may also be effective at the systemic level.

## 2. Results

### 2.1. CpKO Mice Are Overweight Compared to WT Mice and Cp Administration Partially Prevents Body Weight Increase

At 8 months of age, CpKO mice display both systemic and neurological symptoms; therefore, treatment with Cp was done to see the putative therapeutic effect on the overt disease phenotype [24]. Before the Cp administration, at 8 months of age, CpKO mice were overweight as compared to WT mice (Figure 1a).

From 8 to 10 months of age, CpKO mice showed a further 8% increase in body weight (from 38.44 to 41.53 g), while the WT did not (from 29.04 to 29.45 g) (Figure 1a,b). Interestingly, the treatment with Cp showed a trend toward a limitation in body weight increases in the CpKO mice (Figure 1b).

### 2.2. Intraperitoneal Administration Allows Cp to Enter in Both Liver and Adipose Tissue of CpKO Mice

After two months of Cp treatment, we found that Cp protein accumulated in both the liver and perigonadal adipose tissue (pgAT) of CpKO mice (Figure 2a,b). In mouse tissue extracts, the administered human Cp was distinguishable from the endogenous murine Cp by its larger molecular mass (Figure 2, arrows vs. arrowheads), as previously reported [24].

Unexpectedly, the administered human Cp was not detectable in the liver and pgAT of Cp-treated WT mice, suggesting that the exogenous Cp might not accumulate or might be quickly cleared when Cp is physiologically expressed (Figure 2a,b). The comparison of Western blot signals of Cp in the CpKO mice, with that of the purified Cp loaded as control, indicated that Cp accumulated in the different mice in variable amounts, in the range 45–500 pg and 0.5–2.5 ng in 30 µg of total lysate loaded for liver and pgAT, respectively. Although with a wider range, these amounts were in some mice comparable with the estimation obtained for the Western blot signals of the endogenous Cp in WT mice, range 112–447 pg in the liver and from 0.850–1.5 ng in the pgAT. Despite a clear trend of protein’s level reconstitution in CpKO mice, the large variability of rescued levels made not significant the statistical evaluation of the two-ways ANOVA for the interaction of genotype/treatment (Figure 2c,d). These results indicated the retrieval of para-physiological levels of Cp in these organs in the CpKO mice treated with Cp. As mice were perfused before the explant, the detection of Cp signal in the liver and pgAT of CpKO mice treated with Cp was not due to residual blood contained in the organs. Moreover, at the end of the treatment, circulating Cp was not detectable in the serum of these mice [24]. Interestingly, the expression level of the endogenous Cp in WT mice was about 2.1-fold higher in pgAT than in the liver, indicating a large production of Cp by the adipose tissue (Appendix A).

### 2.3. CpKO Mice Show Liver Steatosis, Macrophages Infiltration and IL1ß mRNA Expression Increase Which Are Reduced by Cp Administration

Compared to the WT mice, CpKO mice at 10 months of age exhibited liver steatosis as inferred by Oil Red O staining and by the presence of macrovesicular hepatocytes (Figure 3a–c). However, tissue fibrosis was not evidenced in the CpKO mice (Appendix A). In agreement with the role of Cp in iron homeostasis, lipids accumulation in the liver of CpKO mice was concomitant with iron deposition that was greater than in WT mice, as previously reported for these animals [24].

Quantitative analysis of lipids droplets showed a significant difference between CpKO and WT (genotype effect *p* < 0.0001). Two months of Cp treatment limited liver steatosis in CpKO, reducing accumulation of lipid droplets in hepatocytes (interaction genotype/treatment *p* = 0.0002) (Figure 3a,c). The specificity of the genotype/treatment interaction on CpKO mice was also supported by the lack of effect due to treatment independently by the genotype (treatment effect not significant, ns). The reduction in accumulation of cytosolic lipid droplets was also detected by the reduction of hepatocyte macrovesicles visible in the histological sections (Figure 3b). According to the presence of steatosis, the triglycerides level in the liver of CpKO mice was higher than WT (genotype effect *p* < 0.0001) and showed a trend toward reduction upon Cp administration (interaction genotype/treatment *p* = 0.0896) (Figure 3d). A significant increase of staining for the F4/80 antigen, a marker for Kupffer cells and infiltrating macrophages, was detected in CpKO mice compared to WT mice (genotype effect *p* = 0.0004), which in turn indicated hepatic tissue inflammation (Figure 3e,f). In CpKO liver, the stained cells appeared more interdigitated and widespread than in WT, also showing a more intense staining. The Cp replacement treatment showed a trend in preventing the macrophages infiltration, in the absence of significant genotype/treatment interaction (Figure 3e). However, from the morphological point of view, the F4/80-positive cells in CpKO treated mice were not different from WT (Figure 3f). The inflammatory condition was also indicated by the significant 6-fold increased *IL1ß* mRNA expression in the liver of CpKO mice compared to WT mice (genotype effect *p* < 0.0001). Moreover, CpKO mice treated with Cp showed a significant reduction of the *IL1ß* mRNA (interaction genotype/treatment *p* < 0.0001), indicating a direct effect of Cp in reducing the proinflammatory state (Figure 3g).

### 2.4. Effects of Cp Administration on Iron Accumulation and Expression of Iron Homeostasis-Related Proteins in the Liver

As already reported [24], iron accumulation was observed in the liver of CpKO mice (genotype effect *p* = 0.0080) and was partially prevented by Cp administration, even if the level reached in treated Cp-deficient mice was not significantly different from KO mice (interaction genotype/treatment not significant) (Figure 4a). This partial effect may be due to the late onset of treatment at 8 months of age. We also measured the concentration of copper and zinc metal ions in the liver, but in contrast to iron, they did not show differences between CpKO and WT mice, and their levels were not affected by Cp administration (Appendix A). The peptide hepcidin, the master regulator of iron homeostasis mainly produced by the liver, showed comparable levels in the liver of CpKO and WT mice, in accordance with previous reports [13,25], and was not affected by Cp treatment (Figure 4b).

In mice liver, we also evaluated the expression levels of other proteins directly related to cellular iron homeostasis, namely ferritin (Ft), ferroportin 1 (Fpn) and transferrin receptor 1 (TfR1). Consistent with the observed iron accumulation, we found a significant reduction in expression of Fpn and increase in expression of Ft in CpKO mice compared to WT animals (genotype effect *p* = 0.0128 and *p* = 0.0460, respectively), while TfR1 expression was not different (Figure 4c–f). Treatment with Cp did not change TfR1 and Ft expression in CpKO mice, while iron exporter Fpn expression showed a reduction that was not-significant (interaction genotype/treatment not significant) (Figure 4c–f). In WT mice treated with Cp Fpn and Ft expression levels did not differ from untreated WT animals, and TfR1 level showed a trend toward increased expression suggesting a proper functioning of the iron exporter machinery (Figure 4c–f). Therefore, despite the observed reduction in liver iron accumulation, two months of Cp treatment was not able to fully restore the physiological expression profile of proteins involved in iron homeostasis in CpKO mice.

### 2.5. CpKO Mice Show Adipose Tissue Accumulation and Macrophage Infiltration Reduced by Cp Treatment

CpKO mice of 10 months of age showed increased accumulation of pgAT and subcutaneous adipose tissue (sub-cAT) compared to WT mice (Appendix A). The weight of both the pgAT and sub-cAT collected from CpKO mice were significantly higher than that of WT mice (genotype effect *p* = 0.0382 and *p* = 0.0019 for pgAT and sub-cAT, respectively). The Cp administration was able to reduce the accumulation of adipose tissues in both sites but the interaction genotype/treatment resulted to be significant only for the sub-cAT (*p* = 0.0101) (Figure 5a,b).

To assess whether the increase in adipose tissue mass at 10 months of age in CpKO mice was also paralleled by adipocytes hypertrophy, we measured the size of the adipocytes of pgAT. CpKO mice showed small enlargement of adipocytes size and diameter compared to WT mice, the adipocytes’ diameter showed significant genotype effect (*p* = 0.0338) (Appendix A) suggesting slight hypertrophy of adipocytes in Cp-deficient mice. There was also a significant infiltration of macrophages in the adipose tissue of CpKO mice compared to WT animals (genotype effect *p* < 0.0001), which suggested adipose tissue inflammation (Figure 5c,d; clusters of infiltrating cells are also visible in Appendix A KO panel). Treatment with Cp was efficacious in reducing the infiltration of F4/80-positive cells (a marker of macrophages) (genotype/treatment interaction *p* = 0.0014). A significant effect of the treatment, independently of the genotype, was observed (*p* = 0.0364), even though with opposite effect on CpKO and WT animals. Indeed, while CpKO mice treated with Cp displayed a reduction of the number of infiltrating macrophages, treated WT animals showed increased infiltration of macrophages (Figure 5c). 

### 2.6. In Adipose Tissue of CpKO Mice, Iron Deposition Does Not Occur and the Expression of Iron Homeostasis-Related Proteins Is Comparable to WT Mice

The analysis of the concentration of various metal ions by ICP-MS showed that iron did not accumulate significantly in the pgAT of CpKO mice at 10 months of age compared to WT mice and that the iron level was not affected by Cp replacement (Figure 6a).

Similarly, the copper and zinc concentrations in pgAT was not different among the groups and was not influenced by Cp administration, even though Cu level seemed to be slightly higher in WT mice compared to CpKO, regardless of Cp replacement (Figure 6a). In accord with the evidence of the lack of iron deposition, in pgAT extracts of CpKO mice the expression level of hepcidin was comparable to WT mice and was not significantly affected by Cp treatment (Figure 6b). Similarly, the expression levels of the proteins related to iron homeostasis (TfR1, Fpn and Ft) in CpKO mice were comparable to the levels of the WT animals with no genotype effect (Figure 6c–f). The Cp administration induced an increase of TfR1 expression and a decrease Ft expression, in both WT and CpKO treated mice, demonstrating a conceivable effect of the treatment, while the level of Fpn was not affected (Figure 6c,d,f).

These results indicated that the absence of Cp in the adipose tissue does not create overt derangement in iron homeostasis and that, regardless of the genotype, adipocytes properly respond to the unbalance induced by the presence of exogenous Cp, indirectly indicating the functionality of the administered Cp.

### 2.7. CpKO Mice Show Altered Adipokines Profile in the Adipose Tissue and Increased Circulating Triglycerides That Are Restored by Cp Administration

As iron metabolism seemed not to be affected in the pgAT of CpKO mice, while an inflammatory response was detected, and as Cp is considered an adipokine [17], we evaluated the expression levels of the two major adipokines (adiponectin and leptin) in the pgAT of CpKO mice. These two adipokines are associated with inflammation, lipids metabolism and the cross-talk between adipose tissue and the liver [20,21,22]. Adiponectin expression in the pgAT extracts of CpKO mice was lower than in the WT animals (genotype effect *p* = 0.0452), and the level showed a trend toward an increase by Cp treatment in the CpKO reaching intermediated levels not significantly different from both KO and WT mice (Figure 7a).

Vice versa, leptin expression was higher in the CpKO mice than in WT mice (genotype effect *p* = 0.0042) and showed a trend of reduction upon Cp administration (genotype/treatment interaction *p* = 0.0716) (Figure 7b). These results indicate that the absence of Cp affects the functionality of adipose tissue by altering the profile of secreted adipokines rather than iron homeostasis.

In accordance with altered adipokines profiles, the triglycerides level in the serum of the CpKO mice was also higher than in WT animals (genotype effect *p* = 0.0032). The administration of Cp significantly lowered the level of circulating triglycerides in the CpKO mice (genotype/treatment interaction *p* = 0.0385) becoming similar to that of WT mice (Figure 7c).

### 2.8. Unsupervised Multivariate Analysis of Measured Parameters Distinguishes CpKO from WT Mice and Highlight the Role of Administered Cp in Ameliorating Iron/Lipids Dysmetabolism

Finally, we applied an unsupervised multivariate analysis to further evaluate all the measured parameters related to iron/lipids dysmetabolism alterations at 10 months of age. This type of analysis should highlight whether small differences in different parameters synergize to reveal further insights into the phenotypes of the CpKO mice. 

Principal component analysis was performed using 28 out of 31 parameters quantified in the study (Appendix A); in particular, we excluded the body weight at 8 months of age and the level of Cp in the liver and pgAT that referred to the treatment and not to its effect. The principal component analysis clearly distinguished KO from WT mice. Indeed, a homogeneous cluster of WT mice and WT mice treated with Cp, showing a superimposed 95% confidence interval area, was evidently separated from CpKO mice (Figure 8, cyan and blue dots vs. red dots).

The group of CpKO mice that received Cp treatment was positioned in between the clusters of WT and CpKO mice (Figure 8, green dots), indicating the efficacy of the Cp replacement approach in ameliorating iron/lipids dysmetabolism in the CpKO mice.

## 3. Discussion

This work demonstrates that lack of Cp fosters lipid dysmetabolism in adipose tissue and liver that underline the cross-talk between iron and lipid homeostasis. Lipid dysmetabolism in the CpKO model was evidenced by overweight mice (but not overt obesity) due to adipose tissue accumulation and by slight adipocyte hypertrophy at 10 months of age. Noteworthy was the observation that in these animals, the adipose tissue showed features of inflammation, inferred by macrophage infiltration and by altered profiles of the adipokines, namely a decrease in adiponectin and increased leptin levels as compared to WT mice. Conversely, the lack of Cp expression seems to be less critical for iron homeostasis in adipose tissue. Indeed, CpKO adipocytes did not show iron accumulation or alteration of iron homeostasis-related proteins, which is in line with a previous report [12]. Moreover, the adipose tissue of CpKO mice responds to iron homeostasis imbalance induced by Cp administration by modulating the expression of iron-related proteins as WT mice. This further confirms the activity/functionality of the purified Cp delivered intraperitoneally. Thus, a compensatory mechanism for Cp ferroxidase activity in the CpKO mice might occur in adipocytes due to hephaestin ferroxidase activity [12,26]; indeed, iron accumulation in adipocytes was reported only when both molecules were absent [12].

Our observations and the Cp expression level in the pgAT of WT mice, which is much higher than in the liver, suggest an important role for Cp in the metabolism of adipose tissue in addition to or alternative to cellular iron homeostasis. Cp itself has been defined as adipokine secreted by the adipocytes [17]; thus, it is conceivable that it might contribute to the control of the network of other adipokines released, for example, a metal ions-regulated mechanism. Indeed, Cp is one of the Cu and Zn plasma carriers, and the level of these ions affect lipid metabolism, for example, via zinc-related adipokines [27]. Our results show that Cp is important in pgAT to maintain the physiological levels of adiponectin and leptin, which are essential for the maintenance of energy homeostasis, favoring adipogenesis and lipolysis, respectively, in the adipocytes [28]. This equilibrium, in association with the circulating levels of glucose and insulin, also prevents ectopic lipid accumulation in other organs [28]. In the CpKO mice, as a consequence of the increase in the leptin level, one would expect a reduction of the adipose tissue mass rather than an increase. Nevertheless, in obesity, a decreased sensitivity to leptin occurs; thus, leptin levels are paradoxically increased, likely due to adipose tissue inflammation [29,30]. Indeed, signs of inflammation were evidenced in the CpKO mice that showed macrophage infiltration in adipose tissue in addition to adipokines level imbalance. Pro-inflammatory conditions were previously reported in the adipose tissue of hephaestin/Cp double knockout mice but not in the single CpKO mice, which further did not show variation in adipokines levels [12]. These differences with respect to our observations might be explained by the age of the analyzed mice (6 *vs.* 10 months of age) or might depend on the strategies used to develop the animal models, which were reported to give rise to variable phenotypes, like for example the presence or not of neurological symptoms [5,31,32]. Furthermore, the different methods used for adipokines evaluation (mRNA *vs.* protein) could also be responsible for different results. The use in previous works of mice younger (4–6 months of age) than those used in our work might also explain why differences in body weight of WT and Cp-null mice were not observed [25,33,34]. Indeed, in our ongoing colony of CpKO mice, at 6 months of age, the animals display body weight still comparable to WT mice (data not shown).

The altered levels of adiponectin and leptin found in the pgAT of CpKO mice might reflect an imbalance toward an excessive release by adipose tissue of lipids into circulation, as supported by the increased level of serum triglycerides found in the CpKO mice. The excess of circulating lipids is usually taken up by the liver, which accumulates them, promoting steatosis, and this could be the mechanism occurring in the CpKO mice. Thus, adipokine dysregulation in adipose tissue might contribute, together with iron accumulation, to liver dysmetabolism/damage and diabetes development in the CpKO mice. Indeed, adiponectin is also an insulin-sensitizing adipokine that regulates glucose homeostasis in several tissues, including the liver, where it exerts a protective metabolic effect [28]. Therefore, reduced adiponectin production by the adipose tissue might be detrimental to the liver. Since for the current study we utilized organs collected in a previous investigation, we were unable to assess diabetes-associated parameters, nor, in the case of adipose tissue, were we able to perform qRT-PCR because the stored material was not collected for mRNA analysis and the residual specimens were not enough to yield an adequate amount of mRNA. This is a limitation of our study. 

For the same reason, another limitation is the inability to evaluate the food intake of the mice, a parameter that might have contributed to the body weight increase. We are also aware that the limited number of animals investigated did not allow evaluation of the difference in metabolism that might be associated with sex differences. All these issues deserve future studies in a larger cohort of animals. The generation of a pro-inflammatory environment in the liver contributes to steatosis development [35]. The observed inflammatory conditions in the liver of CpKO mice, underlined by macrophage infiltration and by the increase in IL1ß mRNA levels, the latter already reported together with IL6 [36], might concur to lipids accumulation in the liver, without promoting liver fibrosis, at least in mice of 10 months of age.

In the liver of CpKO mice, we confirmed the important role of Cp in cellular iron homeostasis. In accordance with previous reports [12,13,25,34], the lack of Cp fosters liver iron deposition and an increase in the expression of both Fpn1 and Ft in the absence of changes in hepcidin expression. This important role of Cp in liver iron homeostasis is supported by the reported evidence that hephaestin expression was not able to compensate for the absence of Cp. Therefore, from the iron perspective, the phenotype of CpKO was similar to the double-hephaestin/Cp knockout [12,13]. Liver iron accumulation was always detected in the CpKO mice, as well increase in Ft expression, when measured [5,12,13,31,37]. However, liver steatosis was either not investigated or not reported in these studies. This difference with respect to our results, as was for body weight changes, might depend on differences in age and/or genetic background of the mice investigated.

Adipocyte iron regulates both adiponectin and leptin levels [20,21]; however, we did not observe effects on iron homeostasis in the pgAT of the CpKO mice. Thus, we hypothesize that perturbation of adipose tissue metabolism and adipokines secretion induced by the lack of Cp might be the consequence of the lack of anti-inflammatory/anti-oxidative function of Cp [9]. For example, Cp binds the myeloperoxidase enzyme inhibiting its pro-oxidant functions in inflamed tissues (i.e., adipose tissue and liver) that have been associated with obesity and nonalcoholic steatohepatitis [8,38,39].

Interestingly, some Acp patients show lipid metabolism dysregulation (i.e., overweight, obesity, liver steatosis and high cholesterol), usually considered secondary to diabetes, even if, in some of these cases, diabetes was not diagnosed [14,15,40]. So far, the role of adipocytes in the pathological mechanisms of Acp has not been specifically studied but deserves to be done. Moreover, the alterations in iron and lipids metabolisms observed in the liver and adipose tissue in the absence of Cp may also have implications in other metabolic pathologies like nonalcoholic fatty liver disease characterized by aberrant iron/lipids homeostasis and altered adipokines expression [41,42,43], in which Cp levels have been found to be reduced [44,45,46,47,48].

The potential of Cp-replacement therapy in reducing neurological symptoms was shown in the CpKO mouse model of Acp, providing a new therapeutic perspective [24]. The efficacy of Cp replacement in also ameliorating iron/lipid dysmetabolism in the CpKO mice at the systemic level was highlighted by our multivariate computational analysis. However, despite the observed reduction in liver iron accumulation, steatosis and inflammation, two months of Cp treatment were not fully effective in restoring the physiological profile of the iron homeostasis-related proteins in the liver of CpKO mice. This limited effect might depend on the short treatment duration, not sufficient to affect iron metabolism, and/or on the timing of the treatment (8 months-old mice). Indeed, liver iron deposition is already present at 6 months of age in the CpKO mice [5,31]; thus, earlier treatment might be more efficacious in preventing iron dysmetabolism. Nevertheless, the changes in adipokines levels and macrophage infiltration in the CpKO adipose tissue and the reduction of adipocytes’ hypertrophy indicate that the recovery of Cp level in adipose tissue by Cp treatment is paralleled by the rescue of Cp functionality/role, which resulted efficacious from the therapeutic point of view.

Overall, our results underline the link between iron and lipid dysmetabolism in ceru-loplasmin-deficient mice and indicate ceruloplasmin in adipose tissue could have an anti-inflammatory role rather than its widely known role in iron homeostasis.

## 4. Materials and Methods

### 4.1. Mouse Model and Cp Treatment

Male and female CpKO mice from the original strain (C57Bl/6J genetic background) were used [31]; indeed, no gender differences in Acp penetrance or features have been reported in both human and preclinical models [5,10,31]. Age and sex-matched wild-type (WT) C57Bl/6J mice were used as control. The study was approved by the Institutional Animal Care and Use Committee (IACUC ID 687 and 1010, IRCCS-OSR) and by the National Ministry of Health (n°763/2015-PR and n°629/2019-PR); experiments have been carried out in accordance with EU Directive 2010/63/EU for animal experiments. Mice were from our previous study and their liver and perigonadal adipose tissue and subcutaneous adipose tissue were from −80 °C-stored specimens [24]. CpKO and WT mice of 8 months of age (*n* = 5–6 each group) had been treated for 2 months with human plasma-derived purified Cp (Alexis Biochemicals; 5 μg/g in saline) administered intraperitoneal every 5 days or with saline alone as control [24]. Males and females were distributed to the different groups to match both the CpKO *vs.* WT and the Cp-treated *vs.* untreated groups. At the end of the treatment the blood was drawn, and the mice were euthanized by transcardial perfusion with saline under deep anaesthesia and organs had been collected.

### 4.2. Western Blot Analysis of Protein Expression

Tissues were homogenized in lysis buffer (20 mM Tris, 150 mM NaCl, 1% TritonX100, protease inhibitors), then protein were resolved either on 7.5%- or 10%- or 12%-acrylamide SDS-PAGE and transferred onto a nitrocellulose membrane for Western blot analysis. The antibodies used were: goat anti-Cp (Abcam, Cambridge, UK, ab19171), sheep anti-Cp (Abcam, ab8813), mouse anti-beta actin (Sigma, Milano, Italy, A5441); rabbit anti-H ferritin [49] (donated by Dr. P. Santambrogio, San Raffaele Hospital); mouse anti-transferrin receptor 1 (Invitrogen, Carlsbad, CA, USA, 13-6800); rabbit anti-ferroportin 1 (Invitrogen PA5-64232); and appropriate secondary HRP-conjugated antibodies (Dako, Jena, Germany). Signals were detected using ECL™ reagent (GE-Healthcare, Buckinghamshire, UK) followed by films exposure and image acquisition by the G-Box CCD-camera system (Syngene, Milano, Italy) or by direct image acquisition using a ChemiDoc-MP system (BioRad, Hercules, CA, USA). Densitometric analysis was performed using ImageJ software (U.S. National Institutes of Health, Bethesda, MD, USA), signals were normalized to total protein or to actin expression.

### 4.3. Analysis of Metal Ions by Inductively Coupled Plasma-Mass Spectrometry (ICP-MS)

Iron, copper, and zinc concentrations in the liver and pgAT homogenates were evaluated after digestion (1 h at 70 °C) in a mixture of 65% nitric acid and 30% hydrogen peroxide. The metals content was determined by ICP-MS using an ELAN DRC II instrument (PerkinElmer Sciex, Waltham, MA, USA) as reported [24]. The total quant technique analytical method, with external calibration using a dynamic reaction cell, was adopted. The instrument was calibrated using a standard solution (10 μg/L) (Multielement ICP-MS Calibration Standard 3, PerkinElmer Plus). Each sample underwent two-fold determination. The coefficients of variation ranged from 4.5–7.6% among analytical series and from 5–10.5% between the series. Bovine liver standard (NIST 1640 and MS1577b, National Institute of Standard and Technology, Gaithersburg, MD, USA) was used to better approximate the results from biological matrices. The detection limit, determined on 3 SD of the background signal, was 0.005 μg.

### 4.4. Histological and Immunohistological Analysis

Tissues were fixed in 4% paraformaldehyde (1 h at 4 °C), transferred in 70% ethanol solution and 24 h later embedded in paraffin. Hematoxylin-eosin, Sirius Red staining and immunostaining for the F4/80 antigen detection were performed on a 3 µm-thick section at the Animal Histopathology facility, HSR. Automated immunostaining for macrophage marker F4/80 and cell counterstaining was performed with F4/80 (D2S9R) XP^®^ Rabbit monoclonal antibody (CST#70076, Cell Signaling Technology, Danvers, MA, USA) and hematoxylin, respectively, using the Leica BOND RX instrumentation. Samples were analyzed with a Zeiss AxioImager microscope. F4/80-positive cells or areas on the liver and pgAT immunohistological images were quantified using QuPath software v 0.3.2 [50]; six images from two different tissue sections were acquired for each mouse.

### 4.5. Lipid Droplets Staining

Liver tissues were fixed in 4% paraformaldehyde (12 h at 4 °C), transferred in 30% sucrose solution for 24 h and then embedded in OCT compound (Bio-Optica, Milano, Italy, 05-9801) for storage at −80 °C. Cryostat sections (14 µm-thick) were stained with Lipid Droplets Assay Oil Red O Solution (600045, Cayman Chemical, Ann Arbor, MI, USA) according to the manufacturer’s instructions. Then sections were washed and counterstained with 0.5% Cresyl Violet (Sigma C5042) in 0.3% acetic acid. Samples were analyzed with a Zeiss Axio Imager microscope and images acquired with Nuance FX Multispectral tissue imaging system. For lipid droplet quantitation, images referred to the wavelength of Oil Red O (five images/section) were analyzed with ImageJ software. Nine different areas of the same size were analyzed for each image. The areas were segmented by a threshold filtration to define saturated lipid droplet staining, and then quantified as a percentage of the pixel area covered by lipid droplets on the total the pixel area. Data were reported as average of nine areas × five images × two sections.

### 4.6. Triglyceride Evaluation

Liver and serum triglycerides were determined using the Enzyme Chrome triglyceride assay kit (BioAssay Systems, Hayward, CA, USA). Tissue (10 mg) were homogenized in 100 µL of 5% Triton X-100 and incubated in a water bath for 5 min in which the temperature rose from 80 °C to 100 °C. The procedure was repeated three times, allowing the samples to settle at 20 °C between cycles. Samples were then centrifuged (16,000× *g*, 5 min at 4 °C), the supernatant was diluted 1:10 in Milli-Q water, and 10 µL were used for the assay performed in triplicate in 96-well plates. Sera were diluted 1:5 in Milli-Q water and 10 µL were used for the analysis performed in triplicate in 96-well plates. Samples were mixed with 100 µL of kit working reagent and incubated (15 min at 20 °C) before the plates read at 595 nm wavelength in a BioRad iMark microplate spectrophotometer.

### 4.7. Hepcidin Evaluation

The expression level of hepcidin was evaluated by an ELISA kit (Hepcidin-Mouse #E4693 Biovision, Waltham, MA, USA) in 100 µg of mice liver and perigonadal adipose tissue protein homogenates. Samples were assayed in triplicate in 96-well plates that were read in a BioRad iMark microplate spectrophotometer.

### 4.8. Adipokines Evaluation

Adiponectin and leptin were assessed in pgAT homogenates using ELISA kits (MOB00 and MRP300, R&D Systems, Minneapolis, MN, USA). Samples of adipose tissue homogenates (4.5 µg and 75 µg of protein lysate for adiponectin and leptin, respectively) were assayed in triplicate in 96 well-plates that were read in a BioRad iMark microplate spectrophotometer. The optical density was determined by subtracting the value obtained at 570 nm of wavelength from the value obtained at 450 nm according to the manufacturer’s instructions.

### 4.9. Total RNA Extraction and Quantitative Real-Time PCR

Total RNA was isolated from 5 mg of mouse liver using a Quick-RNA Microprep kit (Zymo Research, Irvine, CA, USA) according to the manufacturer’s protocol. Two µg of total RNA were reverse transcribed using a High-Capacity cDNA Reverse Transcription kit (ThermoFisher, Waltham, MA, USA). The qRT-PCR was performed using the SYBR-Green Master mix (ThermoFisher) according to the manufacturer’s instruction using CFX96 Touch Deep Well Real-Time PCR Detection System (BioRad). The expression of interleukin 1ß (IL1ß) mRNA was measured using the following primers (FW: 5′-TGG AGT TGT ATG CCT CCT ACG-3′; Rev: 5′-AGG CCA CAG GTA TTT TGT CG-3′). Expression level was normalized to that of the housekeeping gene ribosomal protein L13a (Rpl13) mRNA using the following primers (Fw: 5′-AGA AGG GAG ACA GTT CTG CTG-3′; Rev: 5′-TTG CTC GGA TGC CAA AGA GT-3′).

### 4.10. Statistical and Bioinformatics Analysis

For each measured feature, the differences between the four groups of samples were assessed by fitting a two-ways Analysis of Variance (ANOVA) model, considering the main effects of genotype and treatment and their interaction. Overall *p*-values from the two-ways ANOVA models fitted were adjusted using a threshold for false discovery rate ≤ 0.05 (Benjamini and Hochberg method) considering the total number of ANOVA models fitted on each tissue (one model for each feature measured in each tissue; i.e. 14 ANOVA models fitted for liver and 16 ANOVA models fitted for the adipose tissue). The correction was applied separately for the liver and the adipose tissue datasets. Only the models that obtained a significant adjusted *p*-value (FDR < 0.05) were further inspected by considering the individual effects of the regressors (genotype, treatment and their interaction). Comparison of two different groups in Figure 1 was evaluated by unpaired Student’s *t*-test. All the features were preliminary tested for normality using the Kolmogorov-Smirnov test, after an overall Z-score standardization was applied on the full set of samples. All the analyses were performed with Prism V5.0 software (GraphPad Inc.) and the R statistical environment version 3.5.1.

Multivariate analysis, which considers together all the features collected that describe the liver and pgAT pathophysiological state of each mouse (reported in Appendix A), was done using the unsupervised principal component analysis. The data were normalized by auto-scaling transformation (mean-centered and divided by the standard deviation of each variable), and the analysis was performed using MetaboAnalyst 5.0 online package (www.metaboanalyst.ca/home.xhtml) accessed on 18 November 2022. The groups of animals were highlighted as 95% confidence interval area automatedly defined by the software.

## 5. Conclusions

In addition to iron homeostasis, the lack of Cp also dysregulates lipid metabolism, likely due to its potential anti-inflammatory role. This work underlines the cross-talk between liver and adipose tissue in Acp, which in turn suggests that adipocytes may also contribute to the pathology. Moreover, the Cp replacement was efficacious at the systemic level in ameliorating iron and lipid dysmetabolism in the liver and adipose tissue.

## Figures and Tables

**Figure 1 ijms-24-01150-f001:**
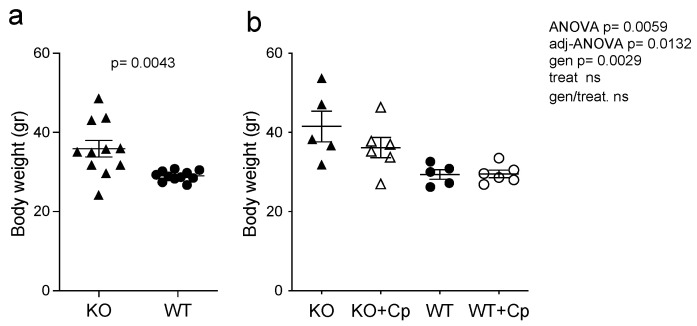
CpKO mice are overweight and Cp administration partially prevents body weight increase. (**a**) Body weight of mice at 8 months of age before the treatment (CpKO and WT, *n* = 11). (**b**) Body weight of mice at 10 months of age after 2 months of treatment with intraperitoneal injection of Cp 5 μg/g, administered every 5 days (KO+Cp and WT+Cp, *n* = 6) or saline (KO and WT, *n* = 5). Data are presented as mean ± SEM of animal groups, each dot corresponds to one mouse. Statistical *p* values were evaluated by Student’s *t*-test in (**a**) and by two-ways ANOVA in (**b**); the latter are reported as value for overall ANOVA, value for Benjamini-Hochberg adjustment for multiple ANOVA tests related to liver features (adj-ANOVA), for genotype variable (gen), for treatment variable (treat) and for the interaction of genotype/treatment (gen/treat).

**Figure 2 ijms-24-01150-f002:**
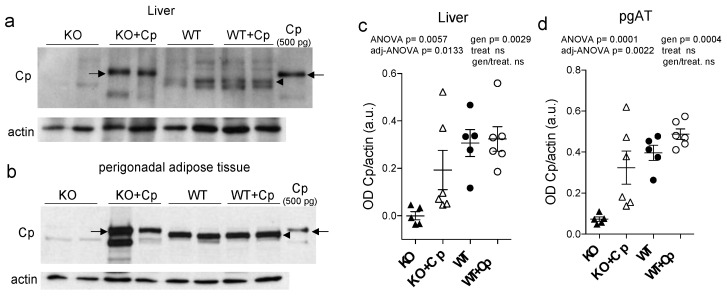
Cp level in liver and perigonadal adipose tissue of CpKO mice treated with Cp. (**a**,**b**) Western blot analysis of Cp and actin levels in liver (**a**) and pgAT (**b**) of mice treated for 2 months with intraperitoneal injection of Cp 5 μg/g, administered every 5 days (KO+Cp; WT+Cp) or saline (KO, WT) (arrow = administered Cp; arrowhead = endogenous Cp); purified Cp was run as control (lanes on the right). Cropped images are from the same or twin SDS-PAGE and Western blot but at different exposure. (**c**,**d**) Densitometric analysis of Cp Western blot signals in liver (**c**) and pgAT (**d**) performed with ImageJ software; values are reported as optical density normalized for actin. Data: mean ± SEM of animal groups, each dot corresponds to one mouse (CpKO, WT *n* = 5; CpKO+Cp, WT+Cp *n* = 6); statistical *p* values were evaluated by two-ways ANOVA and reported as value for overall ANOVA, value for Benjamini-Hochberg adjustment for multiple ANOVA tests related respectively to liver and adipose tissue features (adj-ANOVA), for genotype variable (gen), for treatment variable (treat) and for the interaction of genotype/treatment (gen/treat).

**Figure 3 ijms-24-01150-f003:**
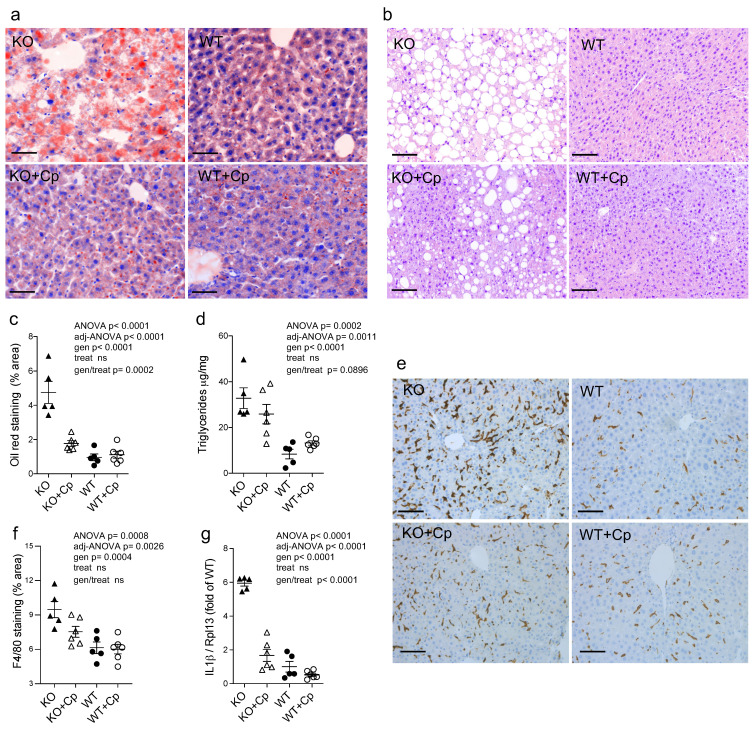
CpKO mice show liver steatosis and macrophage infiltration, which are reduced by Cp administration. (**a**) Oil Red O staining of liver cryostat sections (14 µm-thick) from mice at 10 months of age after 2 months of treatment with an intraperitoneal injection of Cp 5 μg/g, administered every 5 days (KO+Cp; WT+Cp) or saline (KO, WT). (**b**) Histological liver paraffin sections (3 µm-thick) stained by hematoxylin-eosin evidenced of mice as in (**a**). (**c**) Quantitation of lipids droplets determined as a percentage of area occupied by Oil Red O staining in the liver sections of mice treated as in (**a**), images were acquired with Nuance FX Multispectral tissue imaging system and analyzed with ImageJ software. (**d**) Triglycerides concentration in the liver homogenates from mice of different groups was measured by an Enzyme Chrome triglyceride assay kit. (**e**) Staining for the F4/80 macrophages and Kupffer cells marker (brown) in the liver paraffin sections (3 µm-thick); cell nuclei, blue staining. (**f**) Quantitation of the F4/80 staining analyzed using QuPath software. (**g**) Quantitative real-time PCR analysis for *IL1ß* mRNA in the liver of mice; values were normalized to ribosomal protein L13a (*Rpl13*) mRNA expression and reported as a proportion of the WT value. Data: mean ± SEM of animal groups, each dot corresponds to one animal (CpKO, WT *n* = 5; CpKO+Cp, WT+Cp, *n* = 6); statistical *p*-values were evaluated by two-ways ANOVA and reported as value for overall ANOVA, value for Benjamini-Hochberg adjustment for multiple ANOVA tests related to liver features (adj-ANOVA), for genotype variable (gen), for treatment variable (treat) and for the interaction of genotype/treatment (gen/treat). Scale bars = 50 µm in (**a**), and 100 µm in (**b**,**e**).

**Figure 4 ijms-24-01150-f004:**
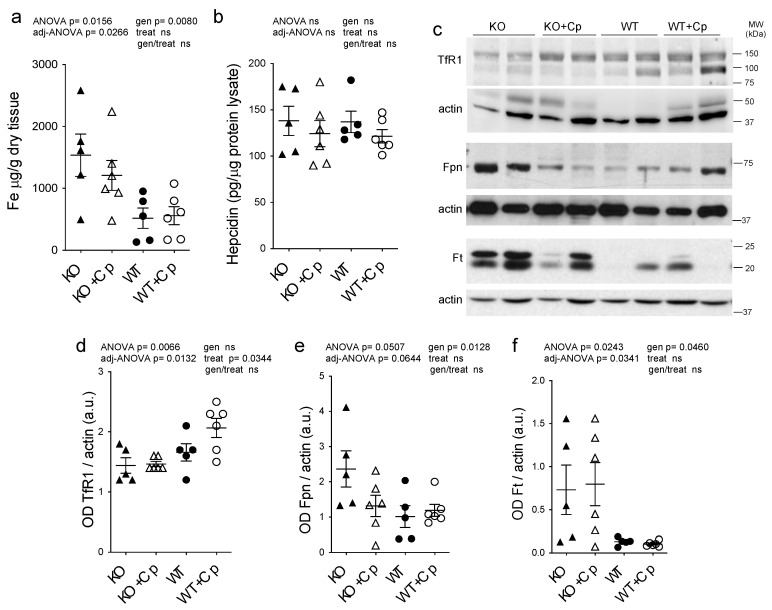
Effects of Cp administration on iron accumulation and iron homeostasis-related proteins expression in the liver. (**a**) Iron evaluation by ICP-MS in the liver of mice after 2 months of treatment with an intraperitoneal injection of Cp 5 μg/g, administered every 5 days (KO+Cp; WT+Cp) or saline (KO, WT). (**b**) Hepcidin level measured by ELISA in the liver lysates of mice treated as in (**a**). (**c**) Western blot analysis of transferrin receptor 1 (TfR1), ferroportin 1 (Fpn), ferritin (Ft) and actin expression in the liver of animals after treatment with Cp (KO+Cp; WT+Cp) or saline (KO, WT). Cropped images are from the same or twin SDS-PAGE and Western blot but at a different exposure. (**d**–**f**) Densitometric analysis of Western blot signals for TfR1 (**d**), Fpn (**e**) and Ft (**f**), performed by ImageJ software; values are reported as optical density (OD) normalized for actin. Data: mean ± SEM of animal groups, each dot corresponds to one animal (CpKO, WT *n* = 5; CpKO+Cp, WT+Cp, *n* = 6); statistical *p*-values were evaluated by two-ways ANOVA and reported as value for overall ANOVA, value for Benjamini-Hochberg adjustment for multiple ANOVA tests related to liver features (adj-ANOVA), for genotype variable (gen), for treatment variable (treat) and for the interaction of genotype/treatment (gen/treat).

**Figure 5 ijms-24-01150-f005:**
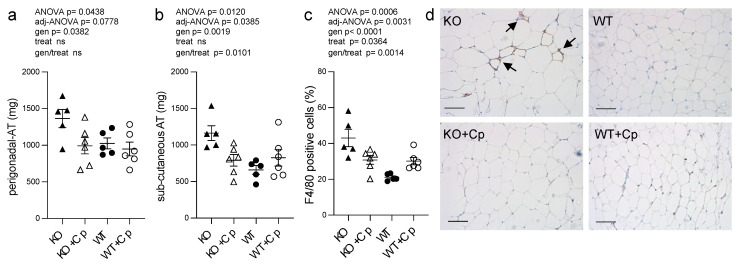
CpKO mice show adipose tissue accumulation and macrophage infiltration, which are reduced by Cp treatment. (**a**) Weight of perigonadal adipose tissue (pgAT) and (**b**) subcutaneous adipose tissue (sub-cAT) collected from mice at 10 months of age after 2 months of treatment with an intraperitoneal injection of Cp 5 μg/g, administered every 5 days (KO+Cp; WT+Cp) or saline (KO, WT). (**c**) Quantitation of F4/80 macrophage marker staining reported as a percentage of positive cells in pgAT, performed with QuPath software. (**d**) Representative immunohistological section of pgAT stained for the F4/80 macrophages marker (brown) and counterstained with hematoxylin (blue). Arrows indicate clusters of infiltrating macrophages. The representative images in panel d were filtered using the “white balance correction 1.0” macro from ImageJ software. Data: mean ± SEM of animal groups, each dot corresponds to one animal (CpKO, WT *n* = 5; CpKO+Cp, WT+Cp, *n* = 6); statistical *p*-values were evaluated by two-ways ANOVA and reported as value for overall ANOVA, value for Benjamini-Hochberg adjustment for multiple ANOVA tests related to adipose tissue features (adj-ANOVA), for genotype variable (gen), for treatment variable (treat) and for the interaction of genotype/treatment (gen/treat). Scale bars = 100 µm.

**Figure 6 ijms-24-01150-f006:**
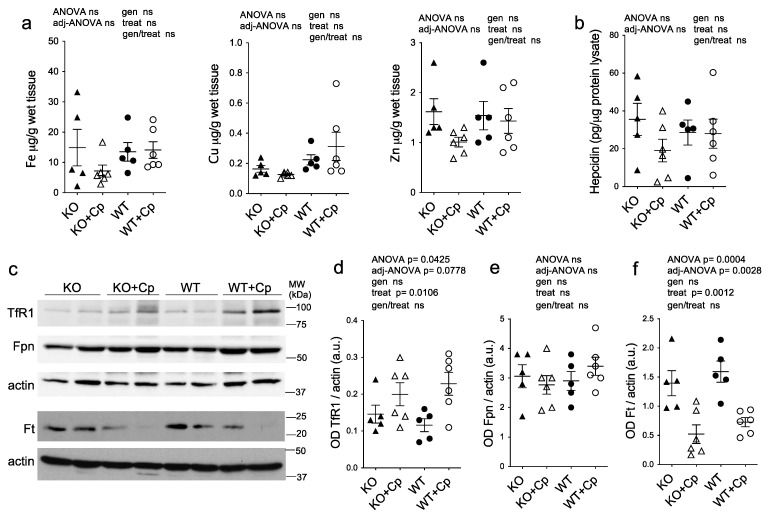
Deposition of metal ions and expression of iron homeostasis-related proteins in perigonadal adipose tissue. (**a**) Iron, copper, and zinc metal ions evaluation by ICP-MS in pgAT from mice at 10 months of age after 2 months of treatment with an intraperitoneal injection of Cp 5 μg/g, administered every 5 days (KO+Cp; WT+Cp) or saline (KO, WT). (**b**) Hepcidin level measured by ELISA in pgAT lysates of mice treated as in (**a**). (**c**) Western blot analysis of transferrin receptor 1 (TfR1), ferroportin 1 (Fpn), ferritin (Ft) and actin expression in pgAT of mice treated as in (**a**). Cropped images are from the same or twin SDS-PAGE and Western blot but at a different exposure. (**d**–**f**) Densitometric analysis of Western blot signals for TfR1(**d**), Fpn (**e**) and Ft (**f**) was performed with ImageJ software; values are reported as optical density (OD) normalized for actin. Data: mean ± SEM of animal groups; each dot corresponds to one animal (CpKO, WT *n* = 5; CpKO+Cp, WT+Cp, *n* = 6); statistical *p*-values were evaluated by two-ways ANOVA and reported as value for overall ANOVA, value for Benjamini-Hochberg adjustment for multiple ANOVA tests related to adipose tissue features (adj-ANOVA), for genotype variable (gen), for treatment variable (treat) and for the interaction of genotype/treatment (gen/treat).

**Figure 7 ijms-24-01150-f007:**
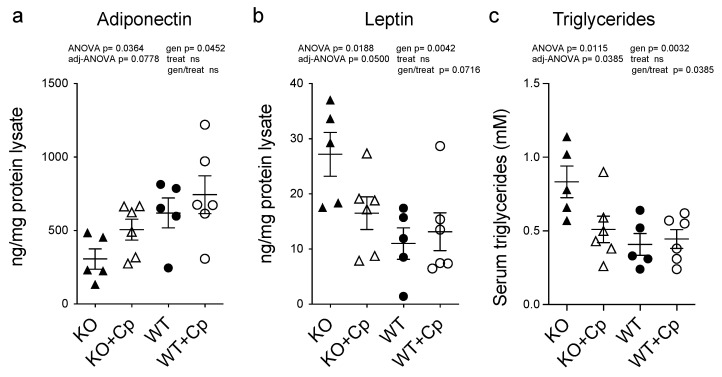
CpKO mice show altered adipokines profile in the adipose tissue and increased circulating triglycerides that are partially restored by Cp administration. (**a**) Adiponectin and (**b**) Leptin concentrations measured by ELISA in pgAT extracts from 10 months-old mice after 2 months of treatment with intraperitoneal injection of Cp 5 μg/g, administered every 5 days (KO+Cp; WT+Cp) or saline (KO, WT). (**c**) Triglycerides concentration in the serum from mice of different groups measured by Enzyme Chrome triglyceride assay kit. Data: mean ± SEM of animal groups, each dot corresponds to one animal (CpKO, WT n = 5; CpKO+Cp, WT+Cp, n = 6). Statistical *p* values were evaluated by two-ways ANOVA and reported as value for overall ANOVA, value for Benjamini-Hochberg adjustment for multiple ANOVA tests related to adipose tissue features (adj-ANOVA), for genotype variable (gen), for treatment variable (treat) and for the interaction of genotype/treatment (gen/treat).

**Figure 8 ijms-24-01150-f008:**
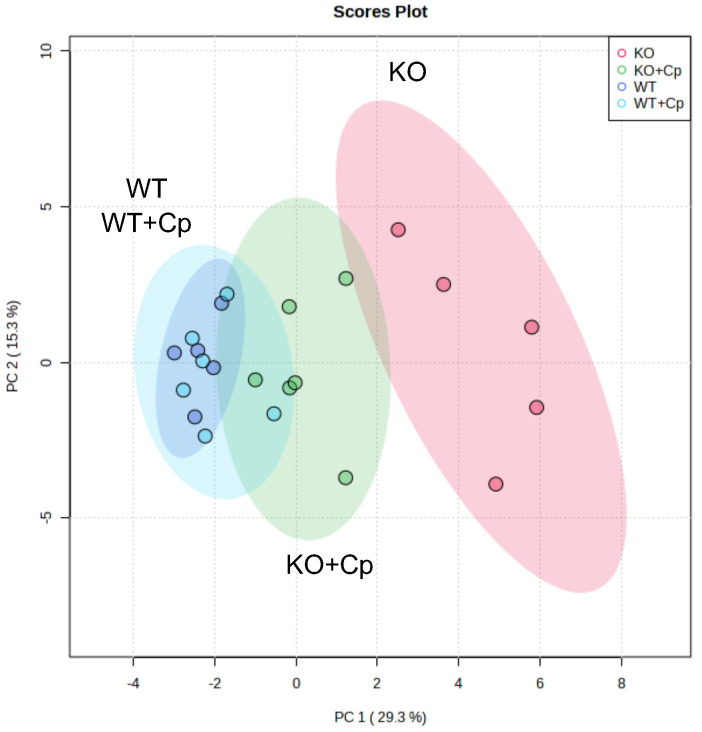
Unsupervised multivariate analysis of measured parameters distinguishes CpKO from WT mice and highlight the role of administered Cp in ameliorating the iron/lipid dysmetabolism. Principal component analysis (PCA) of 28 parameters measured in liver and pgAT. For features see Appendix A. Each dot represents one animal (CpKO = red; WT = blue; CpKO+Cp = green; WT+Cp = cyan); ellipses represent the 95% confidence interval area of each group.

## Data Availability

The datasets of the current study are available from the corresponding author at “San Raffaele Open Research Data Repository” (ORDR) (https://ordr.hsr.it/research-data/).

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
