# Peer review of "Ceruloplasmin-Deficient Mice Show Dysregulation of Lipid Metabolism in Liver and Adipose Tissue Reduced by a Protein Replacement"

_ijms, 2023, doi:10.3390/ijms24021150_

Round 1
Reviewer 1 Report
Raia et all are in the present paper introducing novel function of ceruloplasmin, namely its role in lipid metabolism. This study is a follow up on their previously published paper about ceruloplasmin and its critical role in brain and is excellently supplementing the previous knowledge.
In general, the paper is well written, experiments properly designed and conclussions are also interpretated in a correct way.
The study is actually leaving a good impression and is done with proper controls. However it also opens new exciting questions, which would be nice to either experimentally show or at least discuss.
1. Ceruloplasmin (CP) –KO Mice display overweight, starting at the age of 8 months. Its impressive to see, that even in such “late stage“ the CP treatment can prevent a body increase in these animals. Authors claim (and they are right), that CP plays a critical role in lipid metabolism and observe increase triglycerides, leptin and reduced adiponectin as an mechanism. CP administration is reverting these effects. Any idea how? Is it rather a release of these factors that CP affects or can CP affect the epxression of adiponectin and leptin? Could authors test whether the expression of these factors changes in tissue?
2. Ist also very interesting to see, that the protein is pretty much stable even after 5 days of injection (these observations has been already shown in previous study). However, coming to the liver, could authors show an IHC staining of CP in the liver of KO mice after the treatment?
3. And also whats the sequence of CP used in this study, is it soluble or membrane bound? Please include this info also in the method part.
4. A very minor point. The figure legends could in general contain more information about the experiments. I know they are perfectly mentioned in the material and method part, but for the reader ist just easier to look at the figure and have all the informations directly under it. For example, Figure 1 legend: CP administration, at which concentration? How long? How often injected?,.. etc,..
In general a very nice paper,..
Reviewer 2 Report
The manuscript by Raia and colleagues deals with the effects of ceruloplasmin replacement in ceruloplasmin-deficient mice. The Authors conclude that protein replacement may be effective in rescuing lipid dysmetabolism taking place in the absence of Cp.
In the present form, the data appear too weak to deserve publication. As the authors acknowledge in the last paragraph of Results, “Due to the limited number of animals used and to the variability of the parameters measured, in several analyses we only found trend of differences between groups”. Indeed, too many times differences are not significant to draw conclusions. Their multivariate analysis cannot make up for this lack. I would also add that they have routinely used Student’s t-test (and in one case –I wonder why– Mann Whitney test), while comparison of means of an outcome variable of interest across different groups would require Anova.
The other weakness of the data derives from an incomplete designing of the experiments. Again, the reason emerges from the Authors’ words: “we utilized organs collected in a previous investigation”. Thus, they were forced to analyze tissues already collected for purposes not completely consistent with those of the present study. Even so, there are analyses that could have been done, for instance they claim that Cp deficiency leads to features of inflammation in the adipose tissue as inferred by macrophages infiltration, but do not measure pro-inflammatory cytokines.
There are also more specific points that need attention. As an example, why is the mobility of supplemented Cp different from that of the endogenous protein? It seems that the endogenous Cp is heavily proteolyzed, this should be explained, and the Authors should also clarify how they quantify the protein in this case. They had already observed this phenomenon in a previous paper, but again without giving a convincing explanation.
There are also deficiencies in the bibliographical references and in the English form.
Reviewer 3 Report
Just a side comment: Using Kolmogorov-Smirnov test for small samples has little power and therefore little meaning when sample size is small. Few observations will always point to non-deviation from the normal distribution and therefore accept H0 i.e. normal distribution.
Maybe I did not understand the design correctly but why don't you use paired differences i.e. difference per mouse before and after treatment? Doing this one e.g. could look if there is any regression to the mean effect. An ACOVA like value_treatment=b0+value_baseline+group effect would be most appropriate.
“.. a two-tailed value <0.05 was considered significant comparing means ± standard error.”
I guess you mean comparing means only or standard errors, too? The latter is not the main purpose of ANOVA etc,
You apply several ANOVAS at significance level 0.05. This in turn implies a multiple testing issue across all tests. E.g. in fig 4 the not adjusted p=0.0243 would not be significant if adjustment would have been applied. I strongly suggest to acknowledge this by saying that “… adjustment for number of ANOVAS was not done and therefore p-values should be interpreted exploratorily in that sense.”
The figures show clearly, that KO and WT differ in variability and so ANOVA does too. This may be of little relevance for within group comparisons but between 4-group comparisons the assumption of variance heterogeneity probably is not satisfied.
Line 332: “in particular, were excluded the body weight at 8 months of age ……” should it read “we excluded”?
The PCA analysis result plot looks quite impressive. Still, I wonder how did you choose the number of components?
In conclusion, your many significant p-values show a treatment effect. My main concern apart from multiple testing is why you do not use paired differences thereby reducing variability and not unlikely even “produce” more significant p-values which would work against the multiple testing issue to some extent.
Round 2
Reviewer 2 Report
After my review, the authors performed 1-way ANOVA tests and post-hoc Holm-Šidàk analysis (were ALL pairs compared?) and, as expected, this further slightly lowered the significance rate. They therefore removed sentences were doubts were expressed about the significance of their data, and also moved a figure to the supplementary data section since those data turned out to be not significant. This obviously made the ms less contradictory. I still would love to have a higher number of samples in each group , and see more experiments, as I suggested in my first report.
Please check figure 3, there is some overlap with the old version.
Reviewer 3 Report
Thank you for your response. Just 2 comments again.
1) Thanks for providing p-values for K-S test. Still, the issue I addressed (low power and so accepting H0 even in case of non-normal distribution) persists. It is okay, though to me.
2) Multiple testing and ANOVA: I meant multiple testing because there are more than one ANOVAs involved. So Benjamin-Hochberg, which I assume has been used to correct for multiple testing between pairwise differences within one ANOVA, does not correct for this multiple testing issue between several ANOVAs. Maybe I misunderstood your response otherwise it should be noted even when the results i.e. number of significant p-values do not change.
Author Response
please se attachment
